REGISTERED REPORT PROTOCOL

# Type 1 and type 2 diabetes mellitus: Clinical outcomes due to COVID-19. Protocol of a systematic literature review

**Juan Pablo Pérez Bedoya**[1☯¤]*, **Alejandro Mejía Muñoz**[2☯], **Noël Christopher Barengo**[3‡], **Paula Andrea Diaz Valencia**[1‡]

1 Epidemiology Group, National Faculty of Public Health, University of Antioquia, Medellín, Colombia, 2 Biology and Control of Infectious Diseases Group, Faculty of Exact and Natural Sciences, University of Antioquia, Medellín, Colombia, 3 Department of Translational Medicine, Herbert Wertheim College of Medicine & Department of Global Health, Robert Stempel College of Public Health and Social Work, Florida International University, Miami, FL, United States of America

☯ These authors contributed equally to this work.
¤ Current address: National Faculty of Public Health, University of Antioquia, Medellin, Antioquia, Colombia
‡ NCB and PADV also contributed equally to this work.
* juan.perez42@udea.edu.co

## Abstract

### Introduction

Diabetes has been associated with an increased risk of complications in patients with COVID-19. Most studies do not differentiate between patients with type 1 and type 2 diabetes, which correspond to two pathophysiological distinct diseases that could represent different degrees of clinical compromise.

### Objective

To identify if there are differences in the clinical outcomes of patients with COVID-19 and diabetes (type 1 and type 2) compared to patients with COVID-19 without diabetes.

### Methods

Observational studies of patients with COVID-19 and diabetes (both type 1 and type 2) will be included without restriction of geographic region, gender or age, whose outcome is hospitalization, admission to intensive care unit or mortality compared to patients without diabetes. Two authors will independently perform selection, data extraction, and quality assessment, and a third reviewer will resolve discrepancies. The data will be synthesized regarding the sociodemographic and clinical characteristics of patients with diabetes and without diabetes accompanied by the measure of association for the outcomes. The data will be synthesized regarding the sociodemographic and clinical characteristics of patients with diabetes and without diabetes accompanied by the measure of association for the outcomes.

---

This is a Registered Report and may have an associated publication; please check the article page on the journal site for any related articles.

relevant data from this study will be made available upon study completion.

**Funding:** This research was developed within the framework of the project "Repository for the surveillance of risk factors for chronic diseases in Colombia, the Caribbean and the Americas" and has the financial support of the Ministry of Science, Technology and Innovation of Colombia—Minciencias 844 (grant number 111584467754). The opinions expressed are those of the authors and not necessarily of Minciencias.

**Competing interests:** The authors have declared that no competing interests exist.

## Expected results

Update the evidence regarding the risk of complications in diabetic patients with COVID-19 and in turn synthesize the information available regarding type 1 and type 2 diabetes mellitus, to provide keys to a better understanding of the pathophysiology of diabetics.

## Systematic review registry

This study was registered at the International Prospective Registry for Systematic Reviews (PROSPERO)—CRD42021231942.

## Introduction

The Severe Acute Respiratory Syndrome Coronavirus type 2 (SARS-CoV-2), the causal viral agent of coronavirus disease 2019 (COVID-19), currently has the world in one of the greatest public health crises of recent times since its appearance at the end of 2019 in the city of Wuhan, China [1]. The infection has a mild or even asymptomatic course in most cases, but in elderly patients (over 60 years-of-age) and in those with pre-existing chronic comorbidities, it can progress severe complications such as pneumonia, acute respiratory distress (ARDS) with hyperinflammatory involvement and multi-organ failure, leading in some cases to death [2].

Different studies have reported that patients diagnosed with diabetes who suffer from COVID-19 disease have higher morbidity and mortality compared with people without diabetes [3]. An analysis by Gude Sampedro et al. using prognostic models found that diabetic patients had greater odds of being hospitalized (OR 1.43; 95% CI: 1.18 to 1.73), admitted to the intensive care unit (OR 1.61; 95% CI: 1.12 to 2.31) and dying from COVID-19 (OR 1.79; 95% CI %: 1.38 to 2.32) compared with patients without diabetes [4]. However, it is difficult to establish whether diabetes alone directly contributed to the increase likelihood of complications.

Several studies using secondary data have emerged during the course of the pandemic that seek to determine the association of diabetes with mortality and other clinical outcomes in patients with COVID-19, such as, for example, a meta-analysis carried out by Shang et al. of severe infection and mortality from COVID-19 in diabetic patients compared with those without diabetes. They reported that patients with COVID-19 and diabetes had higher odds of serious infection (OR = 2.38, 95% CI: 2.05 to 2.78) and mortality (OR = 2, 21, 95% CI: 1.83 to 2.66) than patients without diabetes [5]. Despite the fact that there are several primary studies that attempt to explain the association between diabetes and COVID-19, most studies lack epidemiological rigor in the design and methodology used [6]. In addition, many of them did not distinguish between type 1 and type 2 diabetes, which are two very different conditions with different clinical development and pathophysiological mechanisms [7]. This may lead to different degrees of clinical complications from COVID-19. Currently, there is a gap in knowledge about the complications in patients with COVID-19 according to the type of diabetes. Moreover, only limited information exist how COVID-19 affects type 1 patients [8, 9].

The objective of this systematic literature review will be to identify whether there are differences in the clinical outcomes of both type 1 and type 2 diabetes patients diagnosed with COVID-19 compared with patients with COVID-19 without a diagnosis of diabetes. This study will provide scientific evidence regarding the risk of complications in diabetic patients

with COVID-19 and, in turn, synthesize the available information regarding to type 1 and type 2 diabetes.

## Methods

### Study design

This systematic literature review protocol was prepared according to the Preferred Reporting Elements for Systematic Review and Meta-Analysis Protocols (PRISMA-P) [10] (S1 Appendix). The results of the final systematic review will be reported according to the preferred reporting items for systematic reviews and meta-analyses (PRISMA 2020) [11, 12]. In the event of significant deviations from this protocol, they will be reported and published with the results of the review.

### Eligibility criteria

**Participants (population).** Patients with a confirmed diagnosis of COVID-19 without restriction of geographic region, sex, or age. For the diagnosis of COVID-19, the operational definition of confirmed case of the World Health Organization in its latest update will be used as a reference. Confirmed case of SARS-CoV-2 infection: a person with a positive Nucleic Acid Amplification Test (NAAT), regardless of clinical criteria OR epidemiological criteria or a person meeting clinical criteria AND/OR epidemiological criteria (suspect case A) with a positive professional- use or self-test SARS-CoV-2 Antigen RDT [13].

**Exposure.** Patients with COVID-19 and concomitant diagnosis of unspecified diabetes mellitus, differentiated into type 1 diabetes mellitus or type 2 diabetes mellitus, without restriction of geographic region, gender, or age of the patients, who present definition of clinical criteria and /or paraclinical tests used by researchers to classify patients according to their diabetes status.

The operational definition of a confirmed case of diabetes mellitus provided by the American Diabetes Association will be used as a guide. The reference diagnostic criteria for diabetes are fasting plasma glucose ≥126 mg/dL (7.0 mmol/L). Fasting is defined as no caloric intake for at least 8 h or 2-h plasma glucose ≥ 200 mg/dL (11.1 mmol/L) during OGTT or hemoglobin A1C ≥6.5% (48 mmol/mol) or in a patient with classic symptoms of hyperglycemia or hyperglycemic crisis, at random plasma glucose ≥200 mg/dL [14].

In selected primary studies, identification of diabetes status may be based on medical history and International Classification of Diseases codes for type 1 or type 2 diabetes, use of anti-diabetic medications, or previously defined diagnostic criteria.

**Comparator.** Patients with COVID-19 who do not have a concomitant diagnosis of diabetes mellitus.

**Outcome.** The main endpoint is all-cause mortality (according to the definitions of each primary study) and the secondary outcomes are hospitalization and admission to the ICU, where the authors specify a clear definition based on clinical practice guidelines and provide a well-defined criteria for patient outcomes.

**Type of study.** Primary observational original research studies (prospective or retrospective cohort, case-control design, and cross-sectional studies) will be included in this systematic review.

### Exclusion criteria

Clinical trials, editorials, letters to the editor, reviews, case reports, case series, narrative reviews or systematic reviews and meta-analyses, as well as research in the field of basic

sciences based on experimental laboratory models, will be excluded. Original research articles that only include other types of diabetes, such as monogenic diabetes, gestational diabetes, latent autoimmune diabetes in adults, ketosis-prone diabetes, among others, or articles with publication status prior to publication will not be considered. In addition, articles whose main hypothesis is not diabetes and do not have the established outcomes will be excluded.

## Information sources and search strategy

**Electronic bibliographic databases.** For the preparation of the search strategy, the recommendations of the PRISMA-S guide [15] will be adopted. Relevant articles will be identified by electronic search applying the equation previously developed by the researchers and validated by an expert librarian (S2 Appendix). The following electronic bibliographic databases will be used: MEDLINE, EMBASE, LILACS, OVID MEDLINE, WHO (COVID-19 Global literature on coronavirus disease) and SCOPUS with a publication date from December 2019 to August 15, 2022, without language restriction.

The search for potential primary studies published in gray literature will be performed through the World Health Organization database for COVID-19 (WHO COVID-19 Global literature on coronavirus disease). This database contains different electronic bibliographic databases incorporated into its browser, including Web of Science, EuropePMC and Gray literature, among others.

**Unlike electronic bibliographic databases.** To identify other potentially eligible studies, the references of relevant publications will be reviewed to perform a snowball manual search. This technique consists of searching for new articles from the primary studies already selected in order to guarantee exhaustiveness in the search.

## Study selection process

Two researchers will independently evaluate all the titles and abstracts of the retrieved articles, using the free access Rayyan® software [16] with previously established selection criteria. Disagreements will be resolved in first instance through discussion and in the second instance through a third reviewer. Subsequently, the full text of the articles selected in the eligibility phase will be read independently by two researchers, both using the same instrument previously validated in Excel according to predefined criteria. Discrepancies will be resolved by discussion or a third reviewer. The process of identification, selection and inclusion of primary studies will be described and presented using the flowchart recommended by the PRISMA statement in its latest version 2020 [11, 12].

## Data collection and extraction

Standardized and validated forms will be used to collect the data extracted from the primary studies, accompanied by a detailed instruction manual to specify the guiding questions, and avoid the introduction of bias. Data will be extracted from those articles in full text format. If the full text is not available, contact the author or search for the manuscript with the help of the library system. This process will be carried out by two researchers independently. A third investigator will verify the extracted data to ensure the accuracy of the records. The authors of the primary studies will be contacted to resolve any questions that may arise. The reviewers will resolve the disagreements through discussion and one of the two referees will adjudicate the discrepancies presented through discussion and consensus.

In specific terms, the following data will be collected both for the primary studies that report diabetes and COVID-19 and for those that differentiate between DMT1 and DMT2: author, year and country where the study was carried out; study design; general characteristics

of the population, sample size, demographic data of the participants (sex, age, ethnicity), percentage of patients with diabetes, percentage of patients with type 1 and/or type 2 diabetes, percentage of patients without diabetes, frequency of comorbidities in diabetics and non-diabetics, percentage of diabetic and non-diabetic patients who presented the outcomes (hospitalization, ICU admission and mortality) and association measures reported for the outcomes. Data extraction will be done using a Microsoft Excel 365 ® spreadsheets.

## Quality evaluation

The study quality assessment tool provided by the National Institutes of Health (NIH) [17] will be used for observational studies such as cohort, case-control, and cross-sectional. Two tools will be sued: one for cohort and cross-sectional studies (14 questions/domains) and one for case-control studies (12 questions/domains). These tools are aimed at detecting elements that allow evaluation of possible methodological problems, including sources of bias (for example, patient selection, performance, attrition and detection), confounding, study power, the strength of causality in the association between interventions and outcomes, among other factors. The different tools that will be used reflect a score of "1" or "0" depending on the answer "yes" or "no", respectively for each question or domain evaluated, or failing that, the indeterminate criterion option. For observational cohort studies, which consist of 14 risk of bias assessment domains, the studies will be classified as having good quality if they obtain ≥10 points, of fair quality if they obtain 8 to 9 points, and of poor quality if they obtain less than 8 points. On the other hand, in the case of case-control studies that consist of 12 bias risk assessment domains, the studies will be classified as good quality if they obtained ≥8 points, regular quality if they obtained 6 to 7 points and of poor quality if they obtained less than 6 points. However, the internal discussion between the research team will always be considered as the primary quality criterion.

## Data synthesis

A narrative synthesis with summary tables will be carried out according to the recommendations adapted from the Synthesis Without Meta-analysis (SWiM) guide to describe in a structured way the methods used, and the findings found in the primary studies, as well as the criteria for grouping of the studies [18]. A narrative synthesis will be presented in two sections, one for patients with COVID-19 and diabetes and another for patients with COVID-19 and type 1 or type 2 diabetes.

Assessment of clinical and methodological heterogeneity will determine the feasibility of the meta-analysis. Possible sources of heterogeneity identified are the clinical characteristics of the study population, the criteria used to define the outcomes in the groups of patients, the time period of the pandemic in which the study was carried out, and the availability of measurement and control for potential confounding factors. For this reason, it is established a priori that this diversity of findings will make it difficult to carry out an adequate meta-analysis [19]. However, if meta-analysis is considered feasible, the random effects model will be used due to the high probability of heterogeneity between studies. Statistical heterogeneity will be assessed using the $X^2$ test and the $I^2$ statistic, and publication bias assessed using funnel plots if there are sufficient (>10) studies [20].

## Exploratory ecological analysis

An exploratory ecological analysis of the association between the frequency of clinical outcomes of diabetic patients with COVID-19 and the indicators related to the health care dimension, reported for the different countries analyzed by means of the correlation coefficient, will

be carried out. The open public databases of the World Bank (WB) [21], the World Health Organization (WHO) [22] and Our World In Data [23] will be used to extract population indicators related to health care, among those prioritized, universal health coverage, hospital beds per 1,000 people, doctors per 1,000 people, current health spending as a percentage of gross domestic product (GDP), percentage of complete vaccination coverage for COVID-19.

## Discussion

Since the first epidemiological and clinical reports were released from the city of Wuhan regarding the clinical characteristics of patients with COVID-19, a high incidence of chronic non-communicable diseases has been observed in Covid-19 patients. Current scientific evidence has shown that certain comorbidities increase the risk for hospitalization, severity of illness or death from COVID-19, such as hypertension, cardiovascular disease, chronic kidney disease, chronic respiratory disease, diabetes, among others [24].

One of the main chronic comorbidities affected by the COVID-19 pandemic is diabetes. Multivariate analysis of several observational epidemiological studies have revealed that COVID-19 patients with diabetes were at increased risk of hospitalization, ICU admission, and mortality compared with patients without diabetes [4].

For this reason, it is expected that this systematic literature review will provide scientific support regarding the outcomes and complications that patients diagnosed with COVID-19 with type 1 or type 2 diabetes present compared with patients without diabetes. This information will be useful for healthcare personnel, public health professionals and epidemiologists involved in patient care or decision making, generating epidemiological evidence. Thus, highlighting the decisive role of epidemiological research in the context of the pandemic, especially in the field of diabetes epidemiology may improve comprehensive management and care of diabetic patients. This study may also provide important information that can be used to update of clinical practice guidelines.

## Limitations

There are some potential limitations to the proposed systematic review. Firstly, both type 1 and type 2 diabetes may have different key search terms and some studies may be missed. To minimize this limitation, different search equations have been designed for each database in an exhaustive and sensitive manner. In addition to reading references and level ball as an additional strategy. Another limitation is that observational studies evaluating the effect of an intervention may be susceptible to significant confounding bias and may present high heterogeneity in the findings. To report these possible biases, an adequate quality assessment will be carried out, with highly sensitive and previously validated tools, exclusive for each type of observational design. The review is intended for publication in a peer-reviewed journal.

## The status of the study

The study is in the selection phase of the records by applying the eligibility criteria to the titles and abstracts. Completion of the project is expected in September 2022 with the publication of the results.

## Conclusions

This report describes the systematic review protocol that will be utilized to update the evidence regarding the risk of complications in diabetic patients with COVID-19 and in turn synthesize

the information available regarding DM1 and DM2, to provide keys to a better understanding of the pathophysiology of diabetics.

## Supporting information

**S1 Appendix. PRISMA-P (Preferred Reporting Items for Systematic review and Meta-Analysis Protocols) 2015 checklist: Recommended items to address in a systematic review protocol.**
(DOCX)

**S2 Appendix. Search string details for each database.**
(DOCX)

## Author Contributions

**Conceptualization:** Juan Pablo Pérez Bedoya, Alejandro Mejía Muñoz.

**Investigation:** Juan Pablo Pérez Bedoya, Alejandro Mejía Muñoz.

**Methodology:** Juan Pablo Pérez Bedoya, Alejandro Mejía Muñoz.

**Project administration:** Juan Pablo Pérez Bedoya, Alejandro Mejía Muñoz.

**Supervision:** Noël Christopher Barengo, Paula Andrea Diaz Valencia.

**Validation:** Noël Christopher Barengo, Paula Andrea Diaz Valencia.

**Writing – original draft:** Juan Pablo Pérez Bedoya, Alejandro Mejía Muñoz.

**Writing – review & editing:** Noël Christopher Barengo, Paula Andrea Diaz Valencia.

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
