## [Decision Letter · Decision Letter 0]

29 Jul 2022

PONE-D-22-19262Type 1 and type 2 diabetes mellitus: Clinical outcomes due to COVID-19. Protocol of a systematic literature reviewPLOS ONE

Dear Dr. Pérez Bedoya,

Thank you for submitting your manuscript to PLOS ONE. After careful consideration, we feel that it has merit but does not fully meet PLOS ONE’s publication criteria as it currently stands. Therefore, we invite you to submit a revised version of the manuscript that addresses the points raised during the review process. Please submit your revised manuscript by Sep 08 2022 11:59PM. If you will need more time than this to complete your revisions, please reply to this message or contact the journal office at plosone@plos.org. Please include the following items when submitting your revised manuscript:A rebuttal letter that responds to each point raised by the academic editor and reviewer(s). You should upload this letter as a separate file labeled 'Response to Reviewers'.A marked-up copy of your manuscript that highlights changes made to the original version. You should upload this as a separate file labeled 'Revised Manuscript with Track Changes'.An unmarked version of your revised paper without tracked changes. You should upload this as a separate file labeled 'Manuscript'.If applicable, we recommend that you deposit your laboratory protocols in protocols.io to enhance the reproducibility of your results. Protocols.io assigns your protocol its own identifier (DOI) so that it can be cited independently in the future. For instructions see: https://journals.plos.org/plosone/s/submission-guidelines#loc-laboratory-protocols. Additionally, PLOS ONE offers an option for publishing peer-reviewed Lab Protocol articles, which describe protocols hosted on protocols.io. Read more information on sharing protocols at https://plos.org/protocols?utm_medium=editorial-email&utm_source=authorletters&utm_campaign=protocols.

We look forward to receiving your revised manuscript.

Kind regards,

Alok Raghav, PhD

Academic Editor

PLOS ONE

Journal Requirements:

"This research was developed within the "Repository for the surveillance of risk factors for chronic diseases in Colombia, the Caribbean and the Americas" framework and financially supported by a grant of Colciencias 844 (grant number 111584467754)."

"This research was developed within the "Repository for the surveillance of risk factors for chronic diseases in Colombia, the Caribbean and the Americas" framework and financially supported by a grant of Colciencias 844 (grant number 111584467754). The views expressed are those of the authors and not necessarily those of the COLCIENCIAS or the Ministerio de Ciencia Tecnología e Innovación. The funders had and will not have a role in study design, data collection and analysis, decision to publish, or preparation of the manuscript."

Additional Editor Comments:

Authors should address the reviewers comments

Reviewers' comments:

Reviewer's Responses to Questions

**Comments to the Author**

1. Does the manuscript provide a valid rationale for the proposed study, with clearly identified and justified research questions?

Reviewer #1: Yes

Reviewer #2: Yes

2. Is the protocol technically sound and planned in a manner that will lead to a meaningful outcome and allow testing the stated hypotheses?

Reviewer #1: Yes

Reviewer #2: Yes

3. Is the methodology feasible and described in sufficient detail to allow the work to be replicable?

Reviewer #1: Yes

Reviewer #2: Yes

4. Have the authors described where all data underlying the findings will be made available when the study is complete?

Reviewer #1: Yes

Reviewer #2: No

5. Is the manuscript presented in an intelligible fashion and written in standard English?

Reviewer #1: Yes

Reviewer #2: Yes

6. Review Comments to the Author

You may also provide optional suggestions and comments to authors that they might find helpful in planning their study.

Reviewer #1: 1-Operational definition of COVID-19 Diagnosis is required.

2-Please provide diagnostic algorithm as per statement written

3-Operational definition of diagnosis of diabetes mellitus is mandatory

4-Please give heading of snowball manual search and provide details

5-Identify limitation of study and given as separate heading

6-Another similar protocol found "Impact of virtual care on health-related quality of life in children with diabetes mellitus: a systematic review protocol" Justify your study and differentiate between previous studies and protocol.

Reviewer #2: Comments

Overall, this appears to be a well-written systematic review protocol, and the question seems to be worthwhile, with no previous systematic reviews to have comprehensively synthesized the existing evidence related to the effects of COVID-19 on type 2 and type 1 diabetes. The methods for article selection and extraction appear generally sound. The quality of this protocol could be improved by considering the following points;

1.Line 96: delete the word “design”

2.Line 132: it is not clear how the authors will treat studies that had COVID-19 and diabetes co-morbidities but without the outcomes of interest for this systematic review.

3.Authors could consider to search additional databases such as SCOPUS, CINAHL, Web of Science to ensure all potential studies are captured.

4.It was not clear how the authors will treat gray literature, since these often contain useful data that could enhance the quality of systematic review.

5.Line 155-156: how discrepancies between data extractors will be resolved? Authors could choose one method and consistently apply it rather than providing options that may risk reproducibility of the review

6.I'm also curious to know what the authors plan to do if they identify a study without its full-text pdf. Also, I assume the authors will be extracting data from full-text articles, but this could be clearer.

7. PLOS authors have the option to publish the peer review history of their article (what does this mean?). If published, this will include your full peer review and any attached files.

Reviewer #1: **Yes: **Dr. Muhammad Shahzad Aslam

Reviewer #2: **Yes: **Emanuel L. Peter, Ph.D.

---

## [Author Response · Author response to Decision Letter 0]

18 Aug 2022

We appreciate the suggestions made to the submitted manuscript. We make the changes regarding funding, data availability and style requirements. Additionally, we appreciate the corrections made by the reviewers. These corrections were highly relevant to the quality of the protocol. We cover all the recommendations provided

Thank you very much.

---

## [Editor Report · Decision Letter 1]

24 Aug 2022

Type 1 and type 2 diabetes mellitus: Clinical outcomes due to COVID-19. Protocol of a systematic literature review

PONE-D-22-19262R1

Dear Dr. Pérez Bedoya,

We’re pleased to inform you that your manuscript has been judged scientifically suitable for publication and will be formally accepted for publication once it meets all outstanding technical requirements.

Kind regards,

Alok Raghav, PhD

Academic Editor

PLOS ONE
---

## [Editor Report · Acceptance letter]

30 Aug 2022

PONE-D-22-19262R1 

Type 1 and type 2 diabetes mellitus: Clinical outcomes due to COVID-19. Protocol of a systematic literature review. 

Dear Dr. Pérez Bedoya:

I'm pleased to inform you that your manuscript has been deemed suitable for publication in PLOS ONE. Congratulations! Your manuscript is now with our production department. 

Kind regards, 

on behalf of

Dr. Alok Raghav 

Academic Editor

PLOS ONE